# A Critical Review on the Removal and Recovery of Hazardous Cd from Cd-Containing Secondary Resources in Cu-Pb-Zn Smelting Processes

**Guihong Han, Jingwen Wang** , **Hu Sun, Bingbing Liu \*** **and Yanfang Huang \***

School of Chemical Engineering, Zhengzhou University, Zhengzhou 450001, China
* Correspondence: liubingbing@zzu.edu.cn (B.L.); huangyf@zzu.edu.cn (Y.H.)

**Abstract:** Due to the advancement of industrialization and the development of the metal smelting industry, cadmium (Cd), as a highly toxic heavy metal element, is discharged into the natural environment in the form of dust, slag, and waste solutions during the Cu-Pb-Zn smelting process, causing great harm to the soil, water environment, and human health. Meanwhile, Cd is a key component of Ni-Cd batteries and CdTe semiconductor materials. The removal and recovery of Cd from the Cu-Pb-Zn smelting process faces a dual concern with respect to resource recycling and environmental protection. This paper briefly introduces the Cd-containing secondary resources produced in the Cu-Pb-Zn smelting process, systematically reviews the recovery methods of Cd from dust, slag and waste solutions, and compares the technical principles, process parameters, separation efficiency, advantages and disadvantages, and application requirements. In addition, a new route to treat Cd-containing solutions via the foam extraction method was proposed, which has the advantages of a short reaction time, large handling capacity, high removal efficiency, and simple operation equipment, showing superior application prospects, especially for industrial bulk waste solutions with ultralow concentrations.

**Keywords:** cadmium; removal and recovery; separation; secondary resources; Cu-Pb-Zn smelting



## 1. Introduction

Cadmium (Cd) is a highly toxic and carcinogenic heavy metal, and several countries have listed Cd as one of the major metal pollutants needing imperative prevention and control [1–3]. Cd is a nonessential element for the human body, but it often exists in nature as a compound, with a natural background concentration of 0.53 mg/kg in surface soil [4], which does not affect human health under normal environmental conditions. However, with the advances in industrialization and the development of the heavy metal smelting industry, the detrimental effects of Cd on environmental pollution and human health should not be ignored [5]. The natural sources of Cd in soil, water, and crops are mainly geological weathering [6], while anthropogenic sources are usually sewage irrigation, the misuse of pesticides and fertilizers, and heavy metal ore processing and smelting [7–9], especially Cu, Pb, and Zn smelting, which are the main sources of industrial Cd pollution [9]. The heavy metal smelting process produces many dust, slag, and waste solutions containing toxic heavy metals [10,11], which not only cause long-term harm to groundwater and soil, but also enter the human food chain through contaminated water and crops [12,13]. A study of heavy metal pollution in the vicinities of Pb-Zn smelters reported that air, soil, and crops had different degrees of Cd pollution [14,15]. The effects of Cd pollution on the environment and human health are shown in Figure 1. In the list of carcinogens published by the International Agency for Research on Cancer of the World Health Organization (WHO), cadmium and cadmium compounds are included in grade 1 carcinogens. Long-term exposure to Cd may not only produce various toxic effects on human organs, such as the kidneys, liver, and lungs [16], but also increase the risk of cancer death [17].

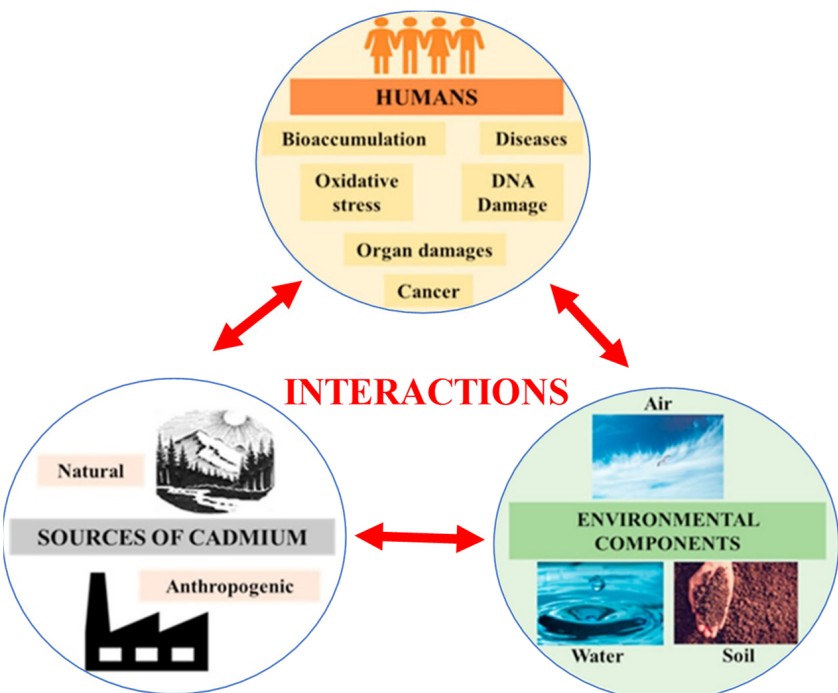

**Figure 1.** The effects of Cd pollution on the environment and human health, reproduced from [18], with permission from Elsevier, 2021.

The geochemical properties of Cd and Zn are very similar, and Cd is usually found in polymetallic sulfide ores, such as sphalerite, smithsonite, Pb-Zn ore, Cu-Pb-Zn ore, and independent greenockite [19,20]. With the rapid consumption of independent Cd ore resources, Cd extraction from the Zn ore, Pb-Zn, and Cu-Pb-Zn ores has received increasing attention. In recent decades, due to the low level of mineral processing technology and equipment, most of the symbiotic or associated Cd resources have not been reasonably utilized [21]. During heavy metal smelting, Cd is mainly concentrated in dust, slag, and waste solution, which not only pollutes the environment but also wastes resources [22,23]. It has been reported that in contaminated soil near heavy metal smelting factories, the depth of soil contamination by Cd reached 4 m, and the average Cd content in the contaminated soil was as high as 74.42 mg/kg, exceeding the Chinese soil environmental quality standard (GB 36,600–2018) [24,25].

Based on the relevant studies at home and abroad, this paper briefly summarizes the characteristics of Cd resources and systematically reviews various methods of separation and recovery of Cd from secondary resources produced in the Cu-Pb-Zn smelting process. The technical principles, process parameters, separation efficiency, advantages and disadvantages, and application requirements were compared. Finally, future research directions for the separation and recovery of Cd are discussed.

## 2. Cd Resources and Cd-Containing Secondary Resources

### 2.1. Cd Resources and Products

Global Cd resources are relatively scarce. According to the data of the USGS, the proven Cd reserves are approximately 500,000 tons, and the Cd reserve distribution is consistent with the distribution tendency of Zn resources. As shown in Figure 2a, approximately 75% of Cd resources are distributed in China, Peru, Russia, Mexico, India, Kazakhstan, the United States, Canada, and Poland [26,27].

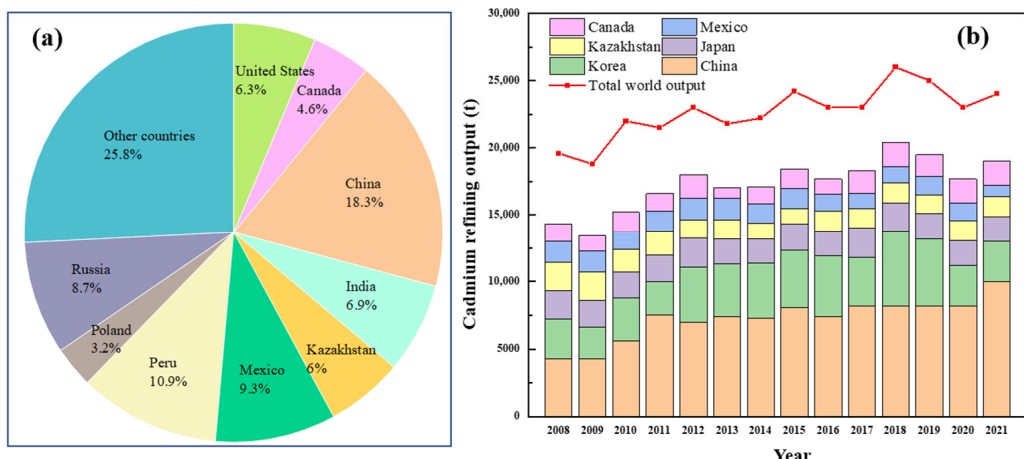

**Figure 2.** (**a**) Cd resource reserves in the world; (**b**) global refined Cd production.

The global annual refined Cd output is approximately 25,000 tons, mainly from Asian countries, such as China, South Korea, and Japan. Global refined Cd production statistics are shown in Figure 2b. Over the past decade, the global refined Cd output has shown an overall upward trend, and China accounts for one-third of the global refined Cd output. The growth of the global refined Cd output is mainly due to the continuous development of Zn smelting capacity [28].

One of the most important applications of Cd and its compounds is the preparation of Ni-Cd batteries. Due to the high toxicity of Cd and the difficulty of recycling Ni-Cd batteries, the consumption of Ni-Cd batteries has declined greatly in recent years. From 2009 to 2016, Chinese Ni-Cd battery exports decreased from 250 million to 70 million, and imports decreased from 150 million to 0.1 million [29]. However, with the rapid development of the solar battery industry, cadmium telluride (CdTe) can be used as an excellent semiconductor material in thin film solar cells, and its manufacturing cost is much lower than that of crystal silicon and other solar cell technologies [30]. First Solar, an American company, is one of the world's largest solar power vendors, and its annual production capacity of CdTe thin film solar cells reached 8 GW in the three branch factories, including the United States, Vietnam, and Malaysia, which is expected to increase to over 16 GW by 2024 [31]. CdTe has a highly stable lattice, which can also show excellent performance under high temperatures or other extreme conditions, showing broad application prospects [32]. The remaining Cd resources are used to produce pigments, coatings, plastic stabilizers, and nonferrous alloys.

Due to the low content and high dispersion of Cd in the Earth's crust, it is not easy to form independent minerals. Cd has geochemical properties and behaviors similar to those of Zn, Ag, Sb, Fe, and other elements, resulting in the isomorphic replacement of these metal cations by Cd [20,33]. Under different geological environments, Cd is mainly sulfurophilic and lithophilic. Most of the Cd resources are endowed with other sulfides in a homogeneous manner, such as sphalerite, galena, and tetrahedrite [21,34–36]. Cd can be found in almost all sphalerite; the Cd content in sphalerite is generally more than $10^{-3}$, while the Cd content in galena is generally $10^{-4}$. In addition to the homogeneous occurrence state, some sulfide ores have a strong adsorption capacity, and Cd exists on the surface of some sulfide ores in an adsorption state [37].

### 2.2. Cd-Containing Secondary Resources in Cu-Pb-Zn Smelting Process

Cu, Pb, and Zn ores are often found in symbiotic or associated forms in nature [38,39]. Global Cu-Pb-Zn resources are abundant, and the global Cu, Pb, and Zn reserves announced by the United States Geological Survey (USGS) are $2.1 \times 10^9$ t, $2 \times 10^9$ t, and $1.9 \times 10^9$ t, respectively, mainly in Australia, China, Chile, and other countries. Global Cu, Pb, and Zn resource mine production in 2021 is $2.1 \times 10^7$ t, $4.3 \times 10^6$ t, and $1.2 \times 10^7$ t respectively, and production is still increasing slightly despite the impact of the COVID-19.

Figure 3a shows the typical Pb smelting process. During the smelting condition of oxygen-rich air, part of PbS is oxidized to PbO, which is enriched in sinter, and then the sinter is further reduced and smelted in a blast furnace, in which crude Pb, Pb slag, and dust are generated. Crude Pb is refined by electrolysis, flue dust can further recover valuable metals, such as Pb, Zn, and Cd [40], and the final slag produced can be stockpiled as general solid waste or sold to building material enterprises after water crushing [41]. The content of Cd in the Pb concentrate is low, but the Cd in the flue dust can be up to over 25% during the Pb smelting process [42].

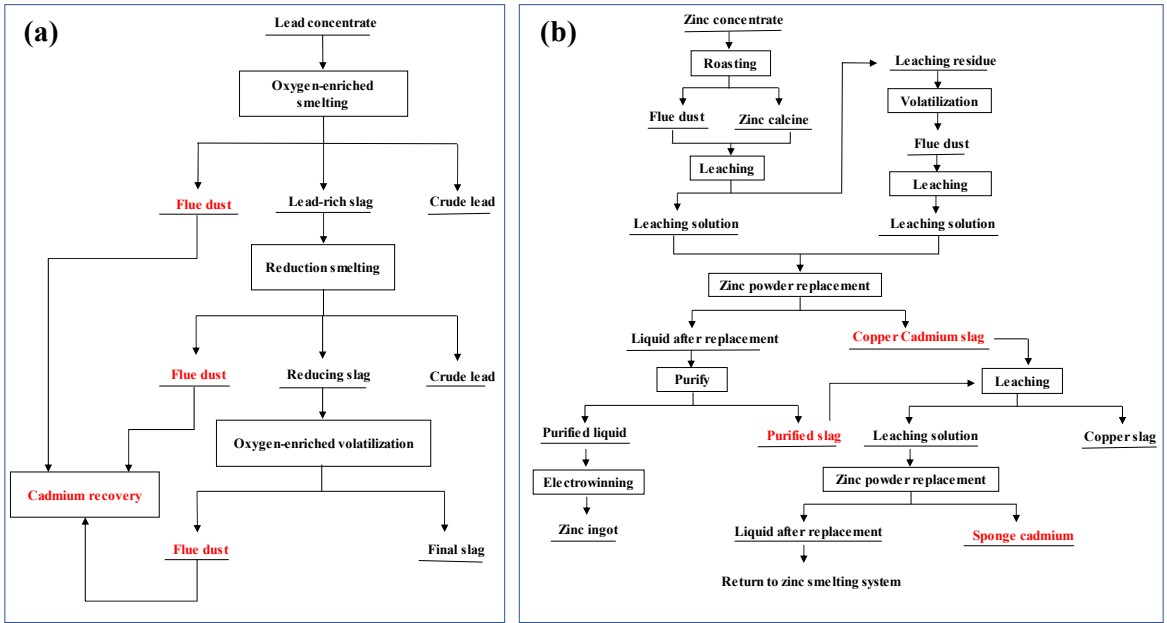

**Figure 3.** (**a**) Pb smelting process; (**b**) Zn smelting process.

Currently, more than 85% of global Zn is produced by the combined process of concentrate roasting—zinc calcine leaching—leach solution purification—purged liquor electrowinning—ingot casting [43]. The Zn smelting process is shown in Figure 3b. During the drying and roasting of the zinc concentrates and the volatile treatment of the leaching slag from zinc calcine, large amounts of Cd-containing dust were produced. Cd exists in Zn concentrate in the form of CdS, which can be transformed into CdO enriched in the dust during oxidation roasting, and this Cd-rich dust can be used to recover Cd. On the other hand, CdS can also be transformed into $CdSO_4$ during sulfation roasting and stayed in the zinc calcine, and the water-soluble $CdSO_4$ is further dissolved into the leachate with the acid leaching of zinc calcine. Some Zn smelters also treated the ZnS concentrates directly by the oxygen pressure acid leaching process, omitting the roasting step, and the CdS was synchronously dissolved into the leachate. The leachate must undergo a purification process to remove impurities, such as Cd, Cu, Co, Mn, In, Ni, etc., to ensure that the Zn electrowinning process is successfully carried out. Cementation can remove Cd, Co, Ni, Cu, and other metals from the solution [44]. The filtered residue obtained from the purification step has a Zn content of approximately 20–50% and a Cd content of approximately 5–10%, which can also be used as a raw material for Cd extraction [45].

In addition, the Cu smelting system also has a certain amount of Cd distribution. Cu smelting is divided into pyrometallurgy and hydrometallurgy. Pyrometallurgy includes roasting, smelting, blowing, and refining processes; hydrometallurgy includes roasting, leaching, and de-hybridization processes [46]. During the roasting of Cu concentrate, volatile impurities, such as Cd, Zn and as in the raw material, can be enriched in the dust, and the Cd content in the dust is approximately 0.03–0.06%. The Cd content in the waste acid produced in the smelting process is approximately 1–150 mg/L, and the Cd content in the rest of the waste solution is approximately 0.1–5 mg/L [47]. The chemical composition

and mineralogical analysis of some dust and slag generated during Cu-Pb-Zn smelting process are summarized in Table 1.

**Table 1.** Chemical compositions and mineralogy analyses of the Cd-containing dusts and slags.

| Types | | Concentration (%) | | | | | | | | | | Mineralogy | Ref. |
|---|---|---|---|---|---|---|---|---|---|---|---|---|---|
| | | Cd | Zn | Pb | Cu | S | As | K | Cl | Co | Mn | | |
| Dust | Zn roasting dust | 18.32 | 0.2 | 42.45 | 0.09 | - | 0.51 | 0.99 | - | - | - | PbSO$_4$, CdO | [48] |
| | Cu smelting open-circuit dust | 1.2 | 2.3 | 21 | 8.6 | 12 | 11.2 | 0.4 | - | - | - | PbSO$_4$, As$_2$O$_3$ | [49] |
| | Pb-Zn smelting Isaar furnace dust | 16.86 | 2.84 | 45.84 | - | 8.67 | 1.4 | - | 1.15 | - | - | PbSO$_4$, CdO, CdS, ZnO | [50] |
| | Secondary Pb materials smelting dust | 13 | 0.42 | 18.5 | - | 3.35 | 6.63 | 5.53 | 11.7 | - | - | KCdCl$_3$, As$_2$O$_3$, CdS, PbClF | [51] |
| | Secondary copper flue dust | 1.27 | 23.4 | 21.8 | 4.67 | 3.32 | 0.16 | 1.8 | 6.21 | - | - | ZnO, PbCl$_2$, Cd$_2$SnO$_4$ | [52] |
| | Pb smelting dust | 18.5 | 3.01 | 18.5 | | 3.35 | 14.1 | 5.53 | 11.7 | - | - | KCdCl$_3$, K$_4$CdCl$_6$, Pb$_5$(AsO$_3$)$_3$Cl | [40] |
| Slag | Zn smelter slag | 4.77 | 44.24 | - | 0.74 | 9.71 | - | - | - | 0.38 | 0.33 | PbSO$_4$, CdO | [53] |
| | Zn neutral leaching residue | 0.26 | 35.99 | 1.73 | 0.52 | 10.05 | 0.41 | - | - | - | 0.74 | ZnFe$_2$O$_4$, ZnO, ZnS | [54] |
| | Zn-Co slag from Zn smelting | 2.57 | 4.9 | - | - | - | - | - | - | 27.8 | 21.47 | ZnO, CdO | [55] |
| | Cu-Cd slag from Zn smelting | 6.43 | 40.9 | 0.99 | 0.98 | 3.35 | 6.63 | - | - | - | - | Zn, ZnO, ZnSO$_4$, Cd, CdO | [56] |
| | Cu-Cd slag from Zn smelting | 21.43 | 28.58 | 1.58 | 1.03 | 8.21 | - | - | 0.03 | - | 0.11 | ZnO, ZnSO$_4$, CdO | [57] |
| | Cu slag from Cu smelting | 0.16 | 4.97 | 0.26 | 1.09 | - | - | - | - | - | 0.09 | FeO$_x$, SiO$_2$, CaO | [58] |

## 3. Cd Recovery Principles and Technologies

### 3.1. Recovery of Cd from Dust

During the smelting processes of Pb and Zn, the roasting of the concentrate and the volatilization of the smelting slag will produce a large amount of Cd-containing dust, which contains approximately 2–25% Cd, mainly in the form of CdO or CdS. At present, most smelters use hydrometallurgical methods to treat Cd-containing dust, mainly including Cd leaching from the dust into the solution followed by the separation and purification of Cd from solution [50]. The Eh-pH diagram of the Cd-S-H$_2$O system and ion species and contents of Cd-H$_2$O system under various pH ranges are shown in Figure 4. It can be seen that Cd$^{2+}$ is stably present in the acidic environment, which is thermodynamically feasible for the dissolution of Cd from the solid waste. The commonly used leaching agents are sulfuric acid and hydrogen peroxide. The main reaction equations are expressed as the following Equations (1)–(3):

$$CdO + H_2SO_4 \rightarrow CdSO_4 + H_2O \tag{1}$$

$$CdS + H_2SO_4 \rightarrow CdSO_4 + H_2S \tag{2}$$

$$CdS + 4H_2O_2 \rightarrow CdSO_4 + 4H_2O \tag{3}$$

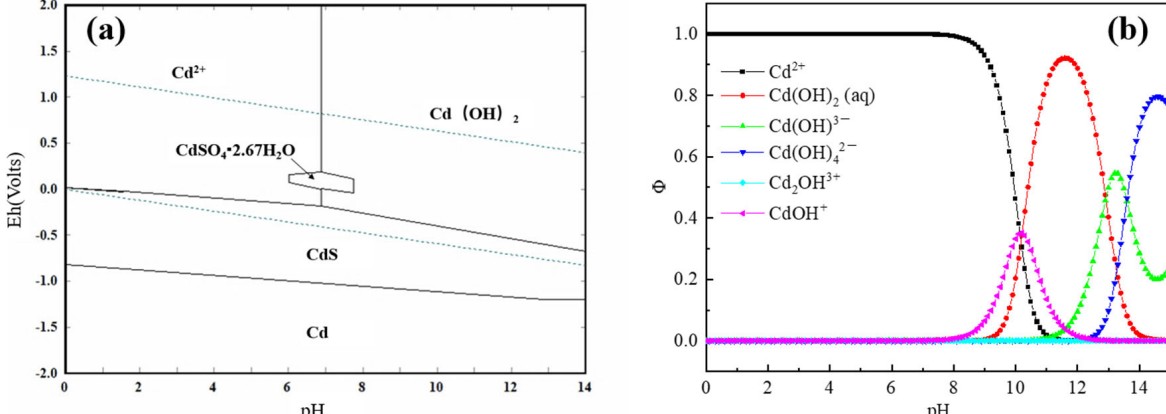

**Figure 4.** (**a**) Eh-pH diagram of the Cd-S-H$_2$O system at 25 °C; (**b**) Ion species and contents of Cd-H$_2$O system under various pH ranges.

### 3.1.1. Sulfuric Acid Leaching

Figure 5 shows the sulfuric acid leaching process of Cd dust produced in the Zn roasting process to prepare high-purity Cd with a microspherical structure. The Cd leaching rate was 95.8% under the conditions of $H_2SO_4$ concentration of 90 g/L, liquid—solid ratio of 6:1 mL/g, leaching temperature of 60 °C, and leaching time of 1 h. Then, the leachate was purified to remove As, Fe, and Cu. Finally, Zn and cadmium were separated using solvent extraction, and cadmium powder with a purity of more than 99.99% was produced by precipitation and hydrogen reduction [59]. An increase in the $H_2SO_4$ concentration can enhance the Cd leaching efficiency. Under the conditions of a $H_2SO_4$ concentration of 110 g/L, liquid—solid ratio of 6:1 mL/g, leaching temperature of 65 °C, and leaching time of 30 min, the Cd leaching rate was increased to 99.63% as the Cd dust with a high content of 18.32% from the zinc volatilized fume dust was leached in sulfuric acid [48]. Waste acid with a Cd concentration of 119 mg/L and a $H_2SO_4$ concentration of 1.7 mol/L produced in the Cu smelting process was used to leach Cd from Cu smelting dust with a Cd content of 1.2%. The Cd leaching rate was as high as 100% under the conditions of a liquid—solid ratio of 4:1 mL/g, leaching temperature of 50 °C, and leaching time of 2 h [49].

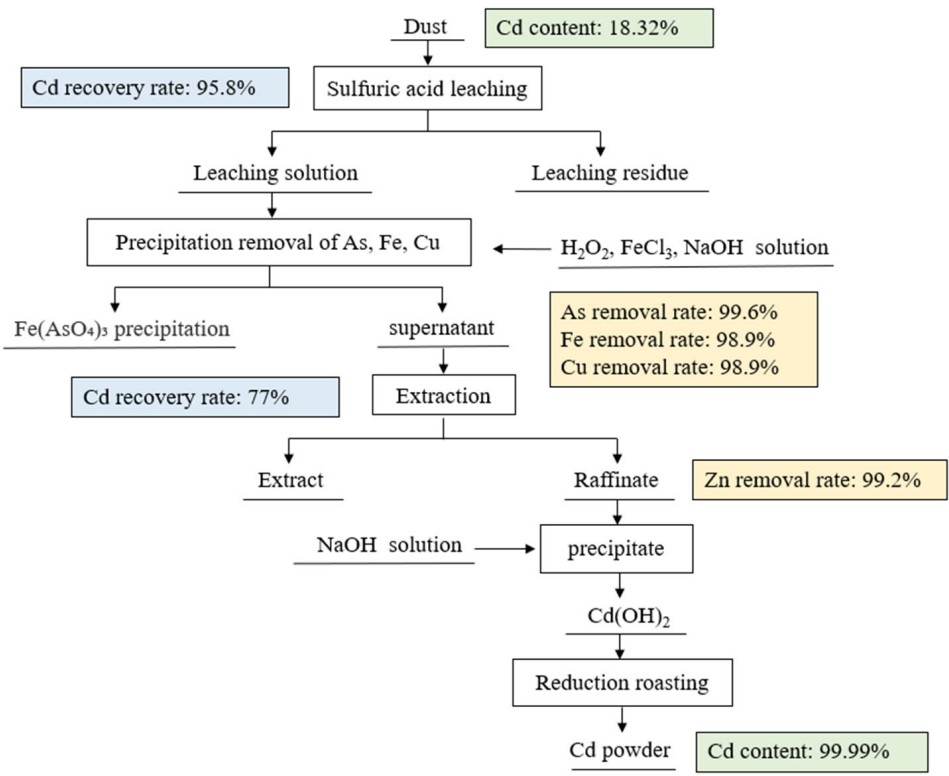

**Figure 5.** High purity Cd was prepared from the Cd-containing dust process by an acid leaching process, reproduced from [59], with permission from Elsevier, 2015.

### 3.1.2. Hydrogen Peroxide Solution Leaching

Hydrogen peroxide ($H_2O_2$) solution can be used as the leaching agent and oxidizing agent for Cd leaching. $H_2O_2$ leaching of As-rich dust with a Cd content of 5.78% produced from the Te smelting process indicated that the Cd leaching rate was 84.3% under the conditions of an $H_2O_2$ dosage of 34.5 mL/L, a liquid-to-solid ratio of 5:1 mL/g, and a leaching time of 5 h [60]. The dust with a Cd content of 16.86% produced by a Pb and Zn smelting enterprise in Yunnan Province, China, was dissolved in a mixed $H_2SO_4$ and $H_2O_2$ solution. Under the optimized conditions of a sulfuric acid concentration of 50 g/L, $H_2O_2$ dosage of 60 g/L, liquid—solid ratio of 3:1 mL/g, leaching temperature of 80 °C, and leaching time of 2 h, the Cd leaching rate reached 95.97% [50]. Moreover, the $H_2O_2$ leaching method is also applicable for the extraction of Cd from recycled secondary Pb

materials with a Cd content of 13%. A Cd leaching rate of 84.2% was achieved when the leaching experiment was conducted at 80 °C for 2 h with a $H_2O_2$ dosage of 100 mL/L [51].

### 3.1.3. Kinetics Study of Cd Leaching

The shrinking core model (SCM) is widely used to describe the dissolution of solid particles in multiphase reaction systems. The leaching process is usually controlled by three kinetic mechanisms: (1) liquid film or solid product layer diffusion control; (2) surface chemical reaction control; (3) mixing control.

The liquid film diffusion control will conform to Equation (4):

$$1 - (1-x)^{2/3} = k_a t \tag{4}$$

The solid product layer diffusion control will conform to Equation (5):

$$1 - \frac{2}{3}x - (1-x)^{2/3} = k_b t \tag{5}$$

The surface chemical reaction control will conform to Equation (6):

$$1 - (1-x)^{1/3} = k_c t \tag{6}$$

The mixing control will conform to Equation (7):

$$\frac{1}{3}\ln(1-x) + (1-x)^{-1/3} - 1 = k_d t \tag{7}$$

where $x$ is the leaching rate of Cd; t is the leaching time; and $k_a$, $k_b$, $k_c$, and $k_d$ represent the apparent rate constants of Equations (4)–(7) [61,62]. Meanwhile, the reaction activation energy can be calculated by Arrhenius equation:

$$k = A exp\left(-\frac{E}{RT}\right) \tag{8}$$

where $k$ and $A$ represent the apparent rate constant and the frequency factor; $E$ and $R$ refer to the activation energy and the universal gas constant. If the activation energy is less than 20 KJ/mol, and the leaching reaction is considered to be controlled by diffusion; if the activation energy value is greater than 42 KJ/mol, it is considered to be controlled by chemical reaction; if it is between 20–42 KJ/mol, it is considered to be obeyed by mixing control [63]. The kinetic analysis of Cd leaching from Cd-containing dust produced in a Zn smelter was performed. The leaching process of Cd was chemically reaction-controlled as judged by the experimental results about the effects of leaching temperature, leaching time, and dust particle size on the leaching rate of Cd. The reaction activation energy was calculated as 51.93 KJ/mol, which is consistent with the chemical reaction control [64].

### 3.2. Recovery of Cd from slag

It has been demonstrated that Cd in metallurgical slag has a great migration ability [65]. In recent decades, most of the slags produced by the Cu-Pb-Zn smelters were directly landfilled without harmless treatment, and the heavy metals in the slags entered the environment through acid rain weathering and leaching, which caused great damage to the soil and underground water [66]. If the slag is directly used as construction material or solidified/stabilized, the valuable metals in the slag are not effectively recycled [36]. Due to the development of industry and a shortage of resources, the resource utilization of slag can promote clean production and resource recycling [67]. The main methods to recover Cd from Cu-Pb-Zn smelting slag include hydrometallurgical processes and bioleaching.

### 3.2.1. Hydrometallurgical Process

The hydrometallurgy process is the most widely used recovery process, which has the advantages of a simple process and low energy consumption, and it can effectively separate different elements selectively from the slag by controlling the leaching conditions. Sulfuric acid leaching of Zn smelting slag from a smelter in Shanxi Province, China, was conducted to separate the valuable metals. The Cd content in the Zn smelting slag was 4.77%, mainly in the form of oxide. Under the conditions of a sulfuric acid concentration of 2 mol/L, liquid—solid ratio of 4:1 mL/g, leaching temperature of 85 °C, and leaching time of 1 h, the leaching rate of Cd reached 85.12%, and the Cd concentration in the leachate was as high as 10.15 g/L. Figure 6 shows the selective separation of valuable metals from zinc smelting slag [53].

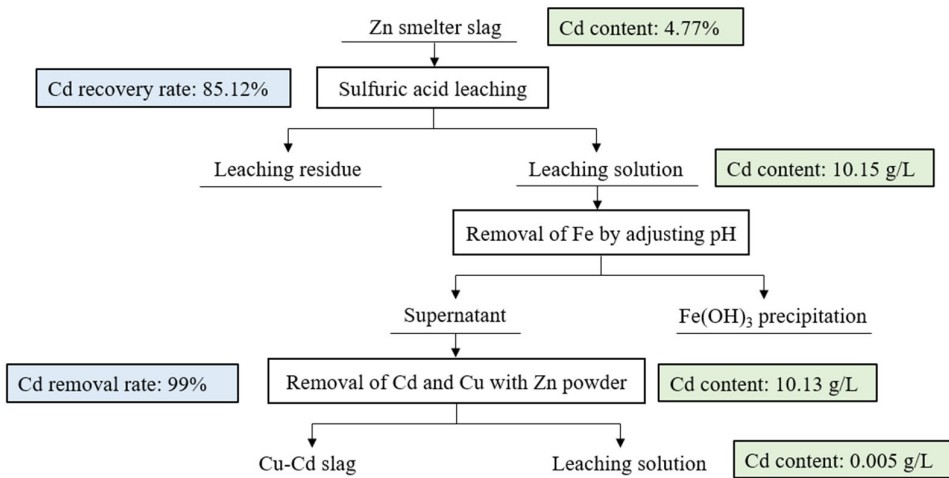

**Figure 6.** Zn Smelting Slag Valuable Metal Recovery Process. reproduced from [53], with permission from Elsevier, 2019.

A Zn leaching slag with a Cd content of 0.26% was obtained from a nonferrous metal smelter in Hunan Province, China. Chemical phase studies have shown that some of the Cd was in the form of Cd-containing Zn ferrite with low solubility. Sulfuric acid was used as the lixiviant, and hydrazine sulfate acted as the reductant to strengthen the reduction of Cd in the acid leaching of Zn slag. The leaching rate of Cd reached 90.81% under the conditions of a liquid—solid ratio of 10:1 mL/g, leaching temperature of 95 °C, and leaching time of 2 h [54].

On the other hand, alkaline leaching can also be used to extract Cd from smelting slag. The purification slag produced in a Zn smelter of Henan Province China was used as raw material, and glycine was used as the leaching agent to selectively leach valuable metals from the slag. The Cd content of the purification slag was 2.57%, mainly in the form of CdO. Under the conditions of a leaching solution pH of 10, glycine concentration of 100 g/L, liquid—solid ratio of 40:1 mL/g, leaching temperature of 45 °C and leaching time of 3 h, the Cd leaching rate reached 87.64%. Combined with the SEM and XPS analyses, it can be seen that most of the glycine amphiphiles were deprotonated, and $Cd^{2+}$ combined with the deprotonated carboxyl functional group in glycine and thus dissolved in the glycine solution [55]. Ammonia leaching of Cu-Cd slag with a Cd content of 6.43% from a wet Zn refinery was investigated to recover Cd. The Cd leaching rate could reach 99.99% with the aid of the oxidant $(NH_4)_2S_2O_8$ under the conditions of an $NH_3 \cdot H_2O$ concentration of 3.7 mol/L, an $NH_4^+$ concentration of 5 mol/L, and an oxidant $(NH_4)_2S_2O_8$ concentration of 30 g/L [56].

### 3.2.2. Bioleaching

Bioleaching is a technique that uses microorganisms to convert valuable metals from ores or slags into soluble elements under mild reaction conditions and is characterized by

low energy consumption and environmental friendliness [68]. A Cu-Cd slag with a Cd content of 11.58% from a smelter in Henan Province, China, was used as raw material for Cd recovery by bioleaching. The flora was isolated, purified, and domesticated to obtain *Bacillus* spp. The Cd leaching rate was 99% at a slurry concentration of 4%, pH of 3, leaching temperature of 30 °C, and leaching time of 5 d [69]. Heterotrophic organisms use expensive yeast extract as energy source, while autotrophic organisms use inexpensive sulfur or pyrite as an energy source, which has greater potential for industrial application. A mixture of acidophilic sulfur-oxidizing bacteria (SOB) and iron ($Fe^{2+}$)-oxidizing bacteria (IOB) was used as bioleaching cells for the bioleaching of Pb-Zn smelting slag. Cd was present in the slag as $CdSO_4$ and Cd $(OH)_2$ and released by acid dissolution. This bioleaching process of Pb-Zn smelting slag has a maximum Cd extraction of 86% [70].

### 3.2.3. Kinetics Study of Cd Leaching

The kinetic analysis of the leaching of Cd from Cd-containing slag was also performed using the SCM. The leaching conditions of Cu-Cd slag generated from Zn smelting process, including leaching temperature, sulfuric acid concentration, slag particle size, liquid-solid ratio, and stirring rate were investigated. The obtained data were tried to fit with the product layer diffusion control and surface chemical reaction control, respectively. It was found that the leaching process of Cd was controlled by the diffusion of ions through the product layer. The activation energy of the reaction was calculated as 10.95 KJ/mol, which is consistent with the diffusion control [71].

The kinetic analysis of the leaching process of Cd from Ni-Cd slag produced during Zn smelting was also carried out. The results showed that the reaction rate was inversely proportional to the square of the particle size, and the leaching of Cd from Ni-Cd slag was also controlled by the diffusion of ions through the product layer. The reaction activation energy was calculated to be 13.36 KJ/mol, which is consistent with the diffusion control [72]. However, kinetic analysis of the common product of Zn smelting, Cd-containing Zn ferrite, showed that, similar to the leaching of Zn and Fe, the leaching of Cd was more fitted to the controlling equations of the chemical reaction. It may be due to the fact that Cd has replaced some of the Zn lattice sites, forming $Cd_{0.25}Zn_{0.75}Fe_2O_4$ and $Cd_{0.5}Zn_{0.5}Fe_2O_4$, while the ferrite has a stable octahedral structure and dissolves more slowly in acidic solutions. The reaction activation energy was calculated to be 79.9 KJ/mol, which is in accordance with the chemical reaction control [73].

### 3.3. Removal and Recovery of Cd from Waste Solutions

Cd exists as $Cd^{2+}$ in waste solutions, and $Cd^{2+}$ is a highly transferable heavy metal ion that is easily dissolved in water or wastewater. According to Chinese current effluent discharge standards, the maximum allowable discharge $Cd^{2+}$ concentration in Class I pollutants is less than 0.1 mg/L. However, the effluent discharged from conventional Zn hydrometallurgical refineries contains approximately 1–4 mg/L of $Cd^{2+}$, and it must undergo a harmless treatment before discharging. Commonly used methods for the removal and recovery of Cd from waste solutions include cementation, precipitation, ion exchange, solvent extraction, adsorption, membrane separation, electrodialysis, electrodeposition, electrocoagulation, and foam flotation [74,75].

### 3.3.1. Cementation

The cementation method is a frequently used process for Zn purification via more positively charged metallic Zn to precipitate and remove $Cd^{2+}$, $Cu^{2+}$, $Co^{2+}$, $Ni^{2+}$, etc., from the solution without introducing new impurities [76–78], and the main principle of the cementation method is shown in Figure 7a. The cementation reaction consists of two steps, including the diffusion step of metal ions and the chemical reaction step. The standard electrode potential difference $\Delta\varphi^0$ between Zn and $Cd^{2+}$ is 0.36 V, and its rate control may belong to diffusion control or mixing control. The main factors affecting the Cd removal rate include the Cd concentration, Zn powder dosage, surface area of Zn powder, temperature,

stirring speed, and pH value. However, Cd often gels on the surface of Zn powder and encapsulates Zn particles [79], and the SEM image of sponge Cd is shown in Figure 7b. In industrial production, to ensure the purity of the electrolyte, it is usually necessary to add an excessive amount of Zn powder when removing $Cd^{2+}$ from the Zn electrolyte [80]. Oxygen in the solution can cause Cd redissolution and affect the Cd removal rate when the reaction temperature is too high or the reaction time is too long [81,82]. Cementation is the most effective method since pure Cd powder can be recovered. However, the appropriate zinc dosage needs to be precisely controlled in industrial operation.

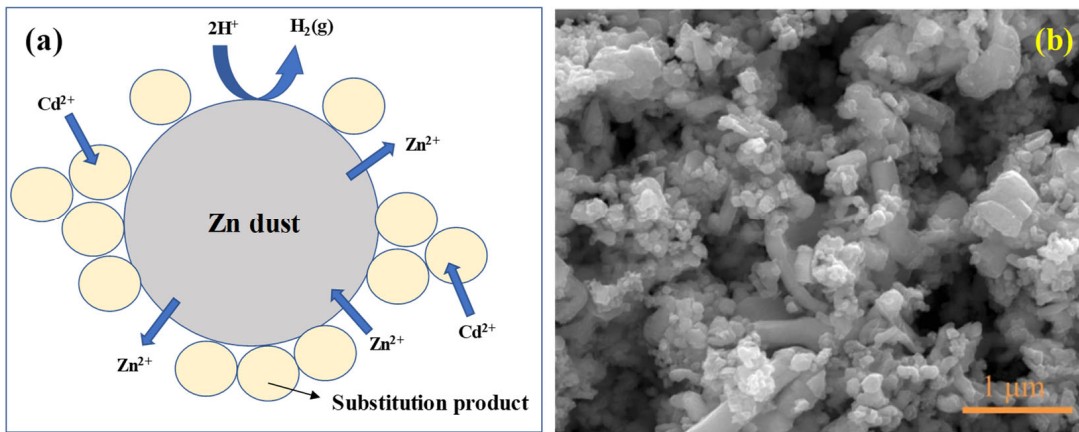

**Figure 7.** (**a**) Principle of the Zn cementation method; (**b**) SEM image of sponge Cd after metallic Zn cementation, reproduced from [83], with permission from Elsevier, 2022.

The graded addition of zinc powder, the reduction of the particle size of zinc powder and the assistance of ultrasonic waves can reduce the Zn dosage required for Cd cementation in industrial electrolytes. For the initial $Cd^{2+}$ concentration in the range of 640–740 mg/L, if Zn powder is added at once, 400% of the theoretical dosage of Zn powder is needed to completely remove $Cd^{2+}$. However, only 200% of the theoretical dosage of Zn powder is needed to achieve 99.9% of Cd removal, as Zn powder is added in three stages [80]. Changing the shape and size of Zn powder can also reduce the required Zn powder dosage. Regarding the initial $Cd^{2+}$ concentration of 400 mg/L, the Cd content of the solution decreased to 18 mg/L in 120 min when the Zn powder had a particle size range of $-100 + 200$ mesh, while the Cd content of the solution decreased to 5 mg/L after 60 min when the Zn powder had a particle size range of $-300$ mesh. Moreover, the surface area of Zn powder particles with a size of $-300$ mesh increased after 1 h of ball milling, and the Cd content of the solution decreased to 1 mg/L after 60 min of reaction [84]. Cd was extracted from the ammoniacal system using an ultrasound-assisted Zn electrode to enhance the replacement of $Cd^{2+}$ by Zn. For the solution with a $Cd^{2+}$ concentration of 20 g/L and $NH_4Cl$ concentration of 50 g/L, the Cd removal rate was 96.68% when the replacement reaction was performed at 35 °C for 6 h with an anodic area ratio of 1:2 and anodic current density of 15 A/m². After introducing the ultrasonic wave with a frequency of 40 kHz, the Cd removal rate was enhanced to 99.21%. The ultrasonic wave not only promoted the separation of Cd from the anode surface and inhibited the generation of floating sponge Cd, but also reduced the concentration polarization of the solution and reduced the region of electrically enhanced cementation reaction potential [79].

### 3.3.2. Precipitation

The precipitation method is used to reduce the number of heavy metal ions in solution by adding precipitants so that the heavy metal ions in solution are combined with the precipitants and transferred to the solid phase. Precipitation removal of $Cd^{2+}$ in solution primarily includes hydroxide precipitation, coprecipitation, carbonate precipitation, and sulfide precipitation [85]. Although the precipitation method is inexpensive and simple

to operate, the precipitation process generates a large amount of hazardous solid waste, which requires further treatment [86].

$Cd^{2+}$ can be converted to water-insoluble $Cd(OH)_2$ by adding an alkali solution to Cd-containing waste solution. Cd-containing waste solution with a Cd concentration of 1.2 g/L from a Zn smelter in China was treated by the hydroxide precipitation method. The removal rate of $Cd^{2+}$ reached more than 90% when the pH value exceeded 10. However, the $Cd(OH)_2$ precipitated particles were suspended and dispersed in solution in water, and flocculants were then added to promote the growth of precipitated particles to facilitate solid—liquid separation. Using $Cd(OH)_2$ as the precipitant and $FeSO_4$ as the flocculant, the average $Cd^{2+}$ concentration in the wastewater after treatment at pH of 11 with a $Cd(OH)_2$ concentration of 10% and $FeSO_4$ concentration of 5% was only 0.086 mg/L, which could achieve standard discharge [87].

The insoluble characteristics of $CdCO_3$ and $CdS$ can be used to remove Cd ions from waste solution. To realize a high removal efficiency of Cd, highly active $Ca(OH)_2$ was used as the precipitant. Eggshells are a common solid waste, and $Ca(OH)_2$ can be prepared by the hydrothermal treatment of eggshells. The synthesized $Ca(OH)_2$ was used to remove $Cd^{2+}$ and $Pb^{2+}$ from wastewater. The effects of $Ca(OH)_2$ dosage on the solution pH and the removal rate of $Cd^{2+}$ and $Pb^{2+}$ are shown in Figure 8a,b. As $Ca(OH)_2$ was added to the solution, the pH value increased rapidly and stabilized at approximately 12. At this pH range, $Cd^{2+}$ in the solution is mainly in the form of $Cd(OH)_2$, which has a certain solubility in acidic or alkaline environments. If gaseous $CO_2$ is introduced into the solution, $Cd(OH)_2$ easily absorbs $CO_2$ and is converted to $CdCO_3$ sediment. Increasing the dosage of $Ca(OH)_2$ can increase the removal rate of $Cd^{2+}$, and the precipitation rate of $Cd^{2+}$ can reach 99.9% under the optimized conditions [88].

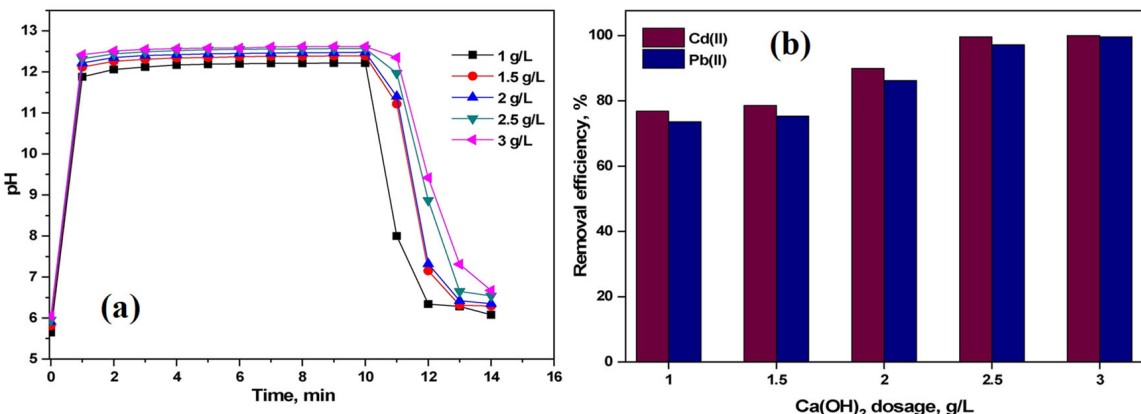

**Figure 8.** (**a**) Changes in pH after adding different concentrations of $Ca(OH)_2$ to simulated wastewater containing $Cd^{2+}$ and $Pb^{2+}$; (**b**) Removal rates of $Cd^{2+}$ and $Pb^{2+}$ at different $Ca(OH)_2$ concentrations, reproduced from [88], with permission from Elsevier, 2020.

With respect to sulfide precipitation, a Cu smelting waste acid with a Cd content of 70.9 mg/L produced by a Cu smelter in Shandong Province, China, was harmless disposed by $H_2S$ synthesized by sulfate-reducing bacteria (SRB). During the precipitation process, $H_2S$ first reacted with $Cu^{2+}$ in the wastewater to produce CuS sediment, and then $H_2S$ started to react with $Cd^{2+}$ to produce CdS when the precipitation of $Cu^{2+}$ was complete, achieving the selective separation of Cd. The final residual $Cd^{2+}$ concentration in the purified wastewater is only 0.08 mg/L, which meets the wastewater discharge standard [86].

The coprecipitation method was also applied to recover the Cd and as from the leachate after hydrogen peroxide leaching of Cd-containing As-Pb dust produced during the smelting of recycled Pb materials. The pH of the leachate was adjusted to 7.5 by NaOH solution, and $Cd^{2+}$ and $As^{5+}$ could be removed simultaneously in the form of $Cd_5H_2(AsO_4)_4 \cdot 4H_2O$ with Cd and As precipitation rates over 99.7% [51].

### 3.3.3. Ion Exchange

To extract or remove the hazardous metal ions from the solution, the ion exchange method refers to the ions in the ion exchange resin being interchanged with the hazardous metal ions in the solution. The ion exchange method is widely used to remove heavy metal ions from wastewater because of its high treatment capacity, high removal efficiency, and short time consumption [89]. Common cation exchange resins include strong acidic resins with sulfonic acid groups (($-SO_3H$) and weak acidic resins with carboxylic acid groups ($-COOH$).

The simulated Cd-containing wastewater can be successfully purified by chelation resin D-401 with iminodiacetate functional groups. To ensure the consistency of the exchangeable ion $Na^+$ content in the resin, the resin was first immersed in HCl solution to replace all $Na^+$ with $H^+$, and then the resin was washed with water followed by immersion in NaOH solution to replace all $H^+$ with $Na^+$, with a target $Na^+$ content of 69 mg/g. Figure 9a,b shows a schematic diagram of the interaction of resin D-401 and $Cd^{2+}$. At a lower pH, large amounts of $H^+$ competed with $Cd^{2+}$ for $Na^+$ on the resin, and the resin's adsorption capacity was reduced. With the increase in pH to 5, the resin had a maximum Cd adsorption capacity of 245 mg/g, and the experimental data were consistent with the Redlich-Peterson isotherm model [90].

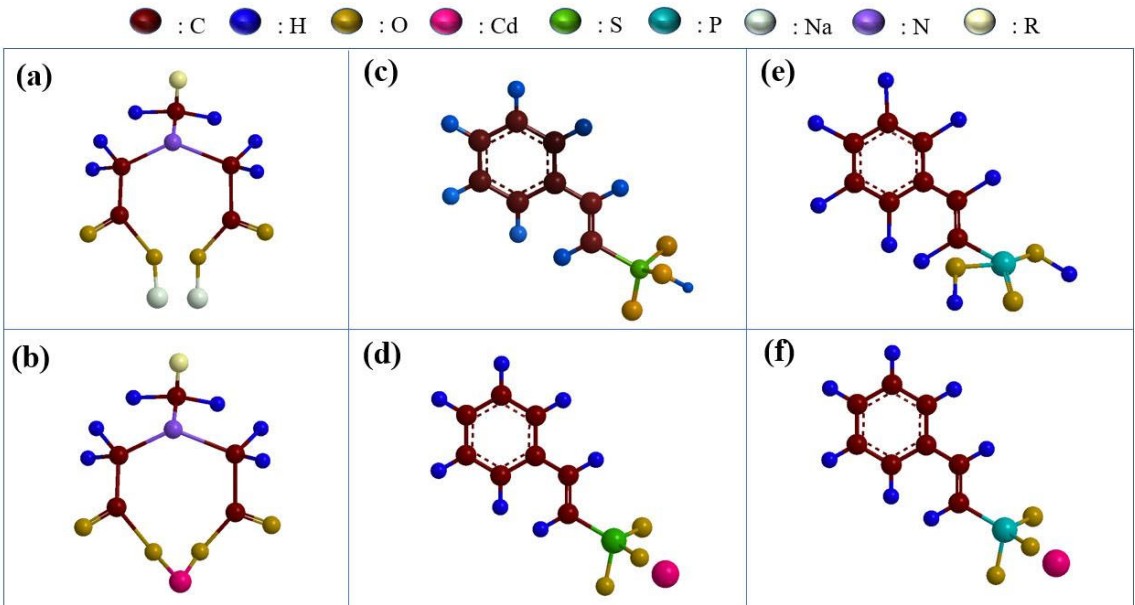

**Figure 9.** (**a**) Molecular structure of D-401; (**b**) molecular structure of the product of $Cd^{2+}$ interacting with D-401; (**c**) molecular structure of 2-phenyl-ethylene sulfonic acid; (**d**) molecular structure of the product of $Cd^{2+}$ interacting with 2-phenyl-ethylene sulfonic acid; (**e**) molecular structure of 2-phenyl-ethylene phosphonic acid; (**f**) molecular structure of the product of $Cd^{2+}$ interacting with 2-phenyl-ethylene phosphonic acid.

Ionic resins containing specific functional groups will have a higher removal rate of $Cd^{2+}$. The adsorption mechanism of $Cd^{2+}$ by Dowex G-26 resin with $-SO_3H$ and Puromet$^{TM}$ MTS9570 resin with both $-SO_3H$ and phosphate ($-PO(OH)_2$) groups was investigated. Figure 9c–f represents the interaction of the two resins with $Cd^{2+}$ in a single structural unit. The O-Cd bonds in the $Cd^{2+}$ binding products with $-SO_3H$ are stronger than those with -$PO(OH)_2$. The free energy change of the $Cd^{2+}$ interaction with most of the structural units containing $-SO_3H$ is negative, while that of the structural units containing $-PO(OH)_2$ is positive. The removal rates of $Cd^{2+}$ from aqueous solutions were 99.68% and 98.95% for G-26 and MTS9570, respectively. Combined with the molecular simulation results and thermodynamic analysis, it was shown that $-SO_3H$ was more effective than $-PO(OH)_2$ in the binding and adsorption of $Cd^{2+}$ [91].

### 3.3.4. Solvent Extraction

Solvent extraction is a unit operation to separate ions in which the components of a system have different solubilities in solvent. Solvent extraction can be used to separate $Cd^{2+}$ and other impurity metal ions. Currently, the commonly used extractants for Cd extraction are D2EHPA [92,93], 2-ethylhexyl phosphate mono-2-ethylhexyl ester [94], TBP [95], Cyanex272 [96], Cyanex923 [97], etc. Solvent extraction is effective for separating $Zn^{2+}$ and $Cd^{2+}$ from solutions [98]. The extraction rate of Cd by different extractants is shown in Figure 10a, and the variation of extraction rate of Zn, Mn, and Cd in chloride solution with different concentrations of D2EHPA at different pH values is shown in Figure 10b. D2EHPA solvent was used as the extractant, and kerosene acted as the diluent to extract $Zn^{2+}$ in chloride solutions containing 5 g/L $Zn^{2+}$, 5 g/L $Cd^{2+}$, and 5 g/L $Mn^{2+}$. The extraction rates of $Zn^{2+}$, $Cd^{2+}$, and $Mn^{2+}$ were 97%, 3%, and 14%, respectively, at a pH of 2.5 and a D2EHPA volume concentration of 10%, showing a satisfactory separation of Zn and Cd [99]. The mixed extractant MDEHPA, consisting of 55% D2EHPA and 45% M2EHPA, was used as the extractant, and toluene was regarded as the diluent to extract Cd from Cd nitrate solution with an initial concentration of 100 mg/L. The Cd extraction rate was 90.9% under the conditions of MDEHPA volume concentration of 6%, solution pH of 6.3, and extraction time of 20 s [100].

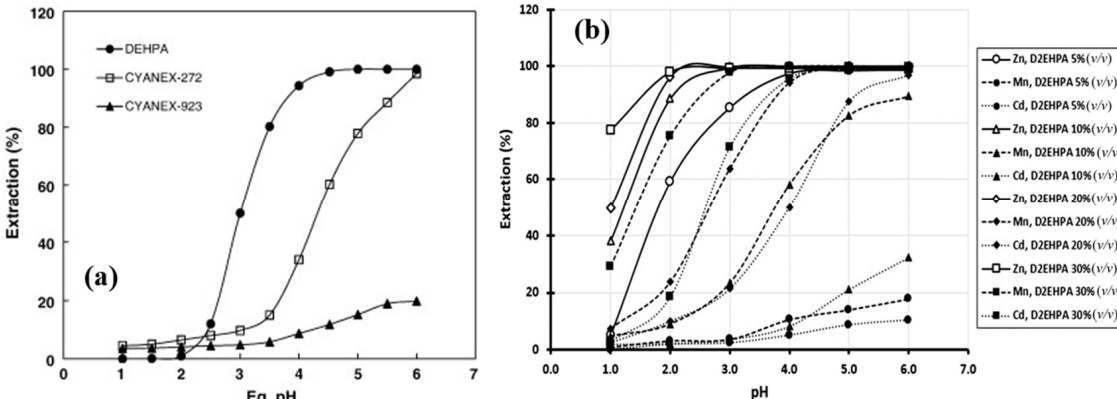

**Figure 10.** (**a**) Extraction rate of Cd by different extractants, reproduced from [101], with permission from Elsevier, 2009; (**b**) Variation of extraction rate of Zn, Mn, and Cd in chloride solution with different concentrations of D2EHPA at different pH values, reproduced from [99], with permission from Elsevier, 2018.

### 3.3.5. Adsorption

Heavy metal ions can be adsorbed on the surface of the adsorbent in the solution to remove hazardous ions. The commonly used kinetic models for the heavy metal adsorption are pseudo-first-order model, pseudo-second-order model, and intraparticle diffusion model. It is reported that the adsorption of $Cd^{2+}$ by most adsorbents conforms to the pseudo-second-order model. The adsorption method has the characteristics of high efficiency, simple operation, wide application, and high removal efficiency [102,103]. Commonly used adsorbents include activated carbon, commercial zeolites, mesoporous materials, etc., but these materials are relatively expensive [104]. In recent years, many high-performance adsorbents, such as biochar, biomodified biochar, graphene oxide (GO), magnetically modified adsorbents, and metal-organic frameworks (MOFs), have been prepared for the adsorption of heavy metal ions in solution.

Biochar is a carbonaceous material prepared by pyrolysis or gasification of biomass materials (such as agricultural waste, wood chips, sludge, animal manure, etc.) [105]. Biochar has a large specific surface area and rich pore structure containing a large number of functional groups, such as hydroxyl, carboxyl, carbonyl, and methyl groups, which have been demonstrated to possess a strong adsorption efficiency [106]. For example, biochar can be prepared from poplar sawdust that was crushed and heated at 600 °C in an inert

atmosphere for 30 min. The prepared biochar could be used to adsorb $Cd^{2+}$, and the involved chemical functional groups, such as carboxyl, amino, and hydroxyl groups, were favorable for the adsorption of $Cd^{2+}$ and $Pb^{2+}$. The principle of adsorption is shown in Figure 11. The adsorption amount of $Cd^{2+}$ was 49.32 mg/g at pH of 5. The $Cd^{2+}$ removal rate of the regenerated biochar obtained after desorption of this biochar was 73.81–80.64% of that of the original biochar, which is reusable. The pseudo-second-order model was adapted to accurately predict the adsorption processes of $Pb^{2+}$ and $Cd^{2+}$ [107].

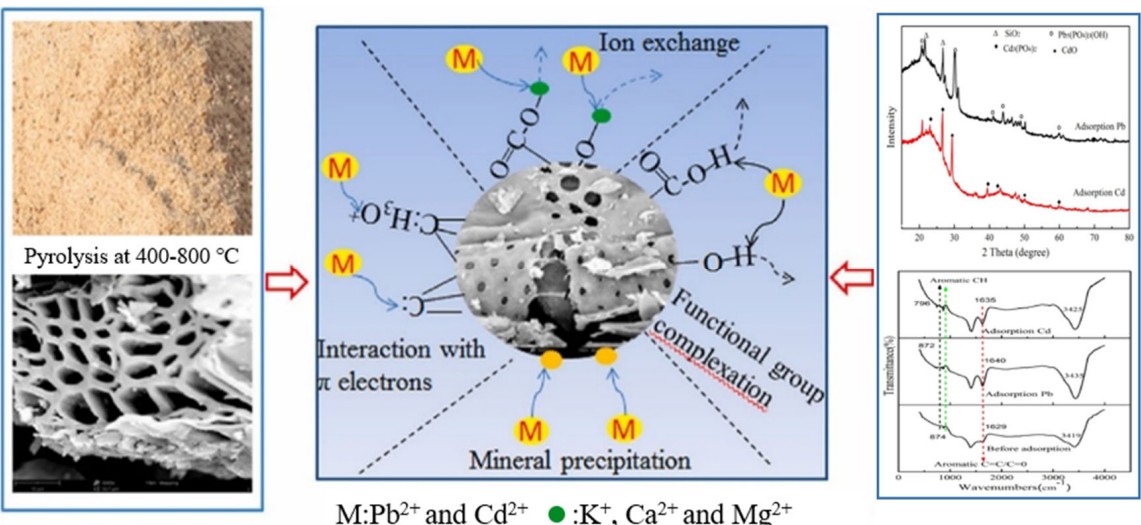

**Figure 11.** Cd and Pb adsorption removal from wastewater by biochar derived from poplar saw, [107], with permission from Elsevier, 2021.

A combination of anaerobic fermentation and pyrolysis processes was used to produce biomodified biochar as a sorbent for $Cd^{2+}$. The unfermented residue left after 24 h of anaerobic fermentation of corn stover was utilized to prepare biomodified biochar (BCB24) via pyrolysis at 500 °C. Compared with the original biochar (CB) produced from corn stover, anaerobic fermentation can significantly increase the surface area, the number of oxygenated functional groups, and the mineral content of the biochar, and the maximum adsorption capacity of BCB24 for $Cd^{2+}$ was 2.1 times higher than that of CB, with an adsorbing capacity of 47.39 mg/g [108].

Graphene oxide (GO) is a product with a large number of oxygen-containing groups, such as carboxyl groups, hydroxyl groups, and epoxy groups, on the surface of the flake layer obtained by the chemical oxidation exfoliation of graphite, and these groups easily chelate with heavy metal ions [109]. It has been reported that the adsorption effect of GO will become poor after the molecules form agglomerates, so GO should be modified to improve its adsorption capacity [110]. A montmorillonite reduced graphene oxide aerogel was prepared by the sol-gel method. Compared with the generally reduced graphene oxide aerogel, the introduction of montmorillonite improved the removal efficiency of $Cd^{2+}$. The maximum static adsorption capacity of $Cd^{2+}$ was 262.79 mg/g, and the desorption efficiency was 80.83% after four cycles. Kinetics calculations showed that the adsorption of M−rGO for Cd was well simulated by the pseudo-second-order kinetic model [111]. The presence of a positively charged octahedral alumina layer between two negatively charged tetrahedral silica sheets in the montmorillonite structure allows montmorillonite and its derived composites to be used for the removal of metal ions from the solution. Serpentine (A), bentonite (B), and iron oxide (I) were used to prepare ternary magnetic ABI composites by a particle—particle coating method, which showed a $Cd^{2+}$ adsorption capacity of 219.2 mg/g and a $Cd^{2+}$ removal rate of 95.1% after the first adsorption cycle and 88.6% after the sixth adsorption cycle. The kinetic model for the adsorption of $Cd^{2+}$ by ABI composites was calculated to be a good fit to the pseudo-second-order kinetic model or the Avrami model [112].

In addition, the doping of silica significantly improved the adsorption capacity of the adsorbent on $Cd^{2+}$. An in-situ hydrolysis method was used to prepare a graphene oxide-silica composite (SGO) for the removal of $Cd^{2+}$ from an aqueous solution. TEM images demonstrated that silica was uniformly distributed on the GO surface. Under the same experimental conditions, the $Cd^{2+}$ removal rates of GO and $SiO_2$ were 66.4% and 28.7%, respectively, while the $Cd^{2+}$ removal rate of SGO for $Cd^{2+}$ was enhanced to 95.87% with a maximum adsorption capacity of 43.45 mg/g. It was calculated that the adsorption kinetic of $Cd^{2+}$ by SGO could fit well with the pseudo-second-order model, which indicated that the adsorption was controlled by chemical reactions [113].

Some adsorbents tend to aggregate in solution and become less effective in practical applications, which can be effectively solved by modifying the adsorbent with magnetic properties [114]. According to hard-soft-acid-base (HSAB) theory, adsorbents rich in carboxyl groups are more suitable for $Cd^{2+}$ adsorption. To this end, a composite adsorbent for the removal of $Cd^{2+}$ from solution was prepared by magnetic modification and carboxyl functionalization. First, magnetic Prussian blue ($Fe_3O_4$@PB) was synthesized by hydrochloric acid etching, followed by surface modification of $Fe_3O_4$@PB using nitrilotriacetic acid (NTA) as a chelation agent to prepare the composite $Fe_3O_4$@PB@NTA. The maximum $Cd^{2+}$ adsorption capacity of this composite adsorbent was 310.56 mg/g, and the removal rate remained at approximately 93.12% after five regenerations. Figure 12a describes the $Cd^{2+}$ adsorption mechanism of $Fe_3O_4$@PB@NTA. It was calculated that the adsorption process could be better described by the pseudo-second-order kinetic model than by the pseudo-first-order kinetic model, which proved that the adsorption of $Cd^{2+}$ by $Fe_3O_4$@PB@NTA was dominated by chemisorption [115]. The chemical composition of the materials before and after adsorption was characterized by X-ray photoelectron spectroscopy, and the following five adsorption mechanisms were deduced: electron sharing between Cd and N provided by O=C-NH and $-C{\equiv}N$, Cd forming O-Cd complexes with O provided by O=CO, Cd sharing electron pairs with O provided by O=CN, and electrostatic attraction force between $COO^-$ and $Cd^{2+}$.

Metal-organic frameworks (MOFs) are constructed by linking metal ions and organic ligands through coordination bonds and have extremely high surface areas and porosities [116]. A new heavy metal ion adsorbent, ZrMOF@GSH, was prepared by immobilizing glutathione as a chelating agent on the surface of the magnetic metal-organic framework. According to the kinetic calculation, the adsorption of ZrMOF@GSH on $Cd^{2+}$ follows the pseudo-second-order adsorption model or intraparticle diffusion model. The maximum adsorption capacity of $Cd^{2+}$ was 393 mg/g, and the removal efficiency of $Cd^{2+}$ was still higher than 90% after three cycles and higher than 80% after ten cycles. The structure of the adsorbent remained intact, indicating that ZrMOF@GSH has good structural stability and reusability [117]. The preparation procedure of $Fe_3O_4$-ZrMOF@GSH and removal of selected heavy metals from the aqueous solution are shown in Figure 12b. A calcium fumarate metal-organic skeleton (CaFuMOF) was prepared as an adsorbent for $Cd^{2+}$ adsorption, and a favorable $Cd^{2+}$ adsorption capacity of 781.2 mg/g was achieved at a solution pH of 7.85. The interaction of the carboxylate anion in the fumarate structure with $Cd^{2+}$ could enhance the central coordination of $Cd^{2+}$. Importantly, CaFuMOF showed good recycling performance. After repeating five adsorption-desorption cycles, the adsorption of $Cd^{2+}$ still reached 80.9%. The adsorption kinetic of $Cd^{2+}$ by the CaFuMOF was calculated to follow the pseudo-first-order adsorption model [118].

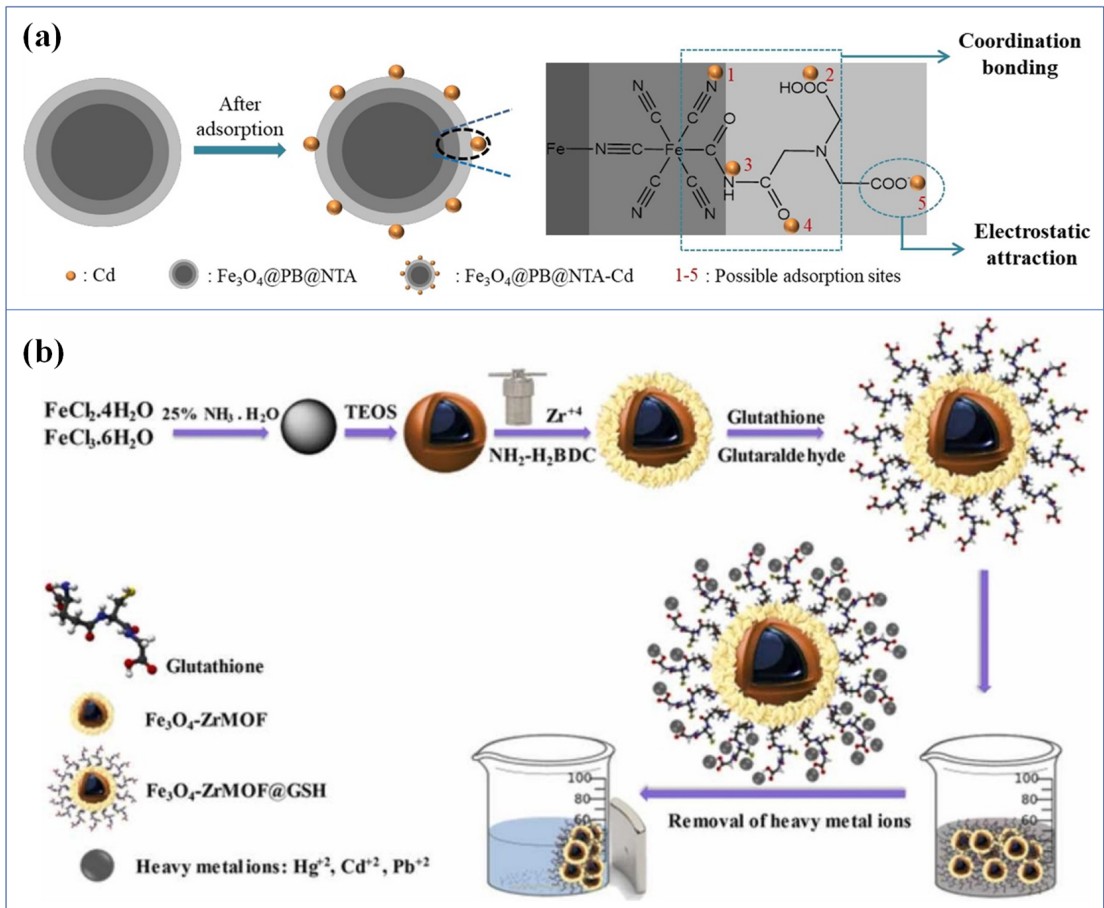

**Figure 12.** (**a**) The adsorption mechanism of Fe$_3$O$_4$@PB@NTA [115], with permission from Elsevier, 2022; (**b**) The preparation procedure of Fe$_3$O$_4$-ZrMOF@GSH and removal of selected heavy metals from aqueous solution reproduced from [117], with permission from Elsevier, 2022.

### 3.3.6. Membrane Separation

The membrane separation method can achieve selective separation of ions with different ionic radii and charged properties from solution [119]. A novel ceramic-polymer composite nanofiltration membrane was prepared using in-situ generated Cu nanoparticles by dip coating on modified ceramic support tubes to remove Cd$^{2+}$ from simulated wastewater with an initial Cd$^{2+}$ concentration of 5 mg/L. The incorporation of nanoparticles transferred the pore size of the modified ceramic base to the nanofiltration membrane range, and a Cd$^{2+}$ removal efficiency of 95.5% was achieved via treatment with the composite membrane under optimal reaction conditions [120].

A Mn-doped hydrous ferrite/cellulose/polyvinyl alcohol (Mn-Fh/Cell/PVA) composite membrane was prepared by loading Mn-doped hydrous ferrite on a cellulose web and wrapped with polyvinyl alcohol for the treatment of simulated As and Cd containing wastewater with a Cd$^{2+}$ initial concentration of 50 mg/L. In the single Cd$^{2+}$ system, the composite membrane had almost no adsorption effect on Cd$^{2+}$. When As$^{3+}$ and Cd$^{2+}$ coexist in wastewater, the electrostatic interaction between them and the composite membrane and the adsorption of the membrane to the Cd-As-OM (metal oxide) ternary surface complex are the mechanisms of As$^{3+}$ and Cd$^{2+}$ co-adsorption by the composite membrane. The adsorption amounts of Cd$^{2+}$ and As$^{3+}$ by the composite membrane were 11.11 mg/g and 72.08 mg/g, respectively [121]. A nanofiber membrane was prepared using polyvinyl alcohol (PVA) and chitosan (Chi) for the treatment of simulated Cd-containing wastewater with an initial Cd$^{2+}$ concentration of 400 mg/L, and the SEM image of the PVA/Chi nanofiber membrane is shown in Figure 13a. The nanofiber membranes were bead-less

with random deposition in the mats, and the fiber diameter was 50–200 nm. The maximum adsorption capacity of the nanofiber membranes for $Cd^{2+}$ was 148 mg/g at a pH of 8 and an adsorption time of 60 min [122].

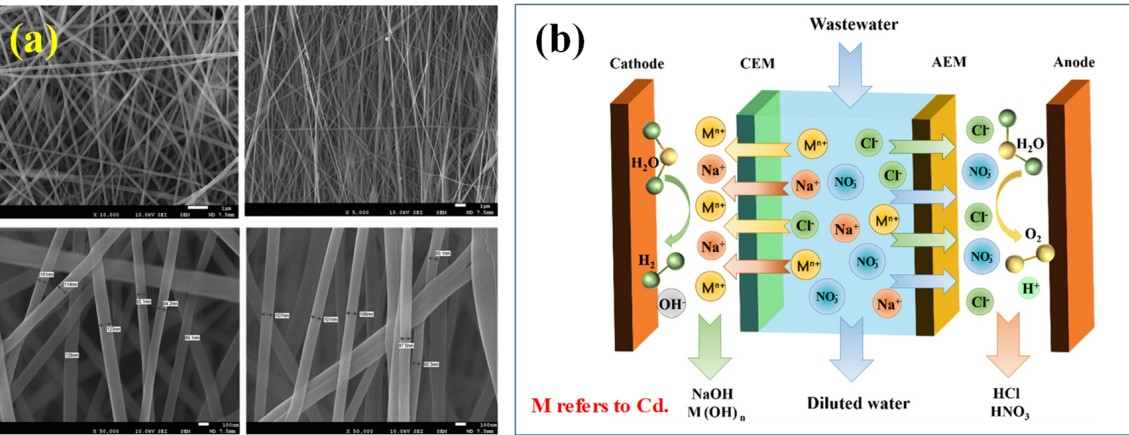

**Figure 13.** (**a**) SEM image of the PVA/Chi nanofiber membrane, reproduced from [122], with permission from Elsevier, 2018; (**b**) Schematic diagram of the electrodialysis process for heavy metal removal [123], with permission from Elsevier, 2022.

3.3.7. Electrodialysis

Electrodialysis is an electromembrane method, in which an anion exchange membrane and a cation exchange membrane are installed between two electrodes. With the potential difference as the driving force, $Cd^{2+}$ can migrate through the cation exchange membrane to the negative electrode, while the anions can cross the anion exchange membrane to the positive electrode, thus achieving the removal of $Cd^{2+}$ from the solution [123,124]. Compared with traditional membrane separation, it has the advantages of fast removal and high removal efficiency under the action of an electric field [125].

A schematic diagram of the electrodialysis process for heavy metal removal is shown in Figure 13b. A three-step bipolar membrane electrodialysis (BMED) system was used to remove As and Cd metals from Cu slag with a Cd concentration of 54.2 mg/kg generated during the copper ore hydrometallurgical process. The heavy metal ions in Cu slag were desorbed in the reaction chamber, and the removal rate of $Cd^{2+}$ was 75.8% after electrodialysis treatment. The pH value of the treated supernatant was 0.77, which could be returned to the Cu hydrometallurgical process as a leaching agent [126].

Ethylenediaminetetraacetic acid (EDTA) is widely used for the selective complexation of various metal ions. A combination of the electrodialysis technique and complexation reaction was used for the separation of Cu and Cd from the solution. In the pH range of 5.0–6.4, Cu can complex with EDTA as an anionic complex ($CuL^{2-}$), while Cd exists as a free ion. $CuL^{2-}$ crossed the anion exchange membrane to the positive electrode, and $Cd^{2+}$ passed through the cation exchange membrane to the negative electrode. Both the concentrations of Cu and Cd ions were lower than 0.3 mg/L after treatment for 90 min, and Cu was not observed in the Cd concentrate near the negative electrode, achieving the selective separation of Cd from Cu [127].

3.3.8. Electrodeposition

During the electrodeposition process, the reduction reaction occurs at the cathode, whereas the oxidation reaction occurs at the anode, in which the cationic metal is deposited on the surface of the cathode. Compared with other methods, the electrodeposition method is simpler and more environmentally friendly, and the purity of the products is higher so that the Cd resources can be recovered directly from the wastewater [128]. Electrodeposition is a liquid-solid reaction. If the liquid-phase mass transfer process is intensified and the concentration polarization decreases, then it can be assumed that the polarization of the

whole electrode is caused by the electrochemical polarization, and the integrated rate of the electrode reaction will be mainly determined by the electrochemical reaction rate. If the liquid-phase mass transfer process is slower and the concentration polarization increases, the rate of the electrode reaction will be determined by the diffusion rate. The main electrode reaction equations are expressed as Equations (9) and (10). However, when the $Cd^{2+}$ concentration is low, the electrodeposition method cannot completely remove $Cd^{2+}$ because hydrogen precipitation reactions and co-deposition of other metals occur during the continuous electrolysis process, which occupies the active sites of the electrodes and thus affects the reduction of $Cd^{2+}$.

$$\text{Cathode: } Cd^{2+} + 2e^- \rightarrow Cd \; 2H_2O + 2e^- \rightarrow H_2 + 2OH^- \tag{9}$$

$$\text{Anode: } 2H_2O \rightarrow O_2 + 4H^+ + 4e^- \tag{10}$$

Simulated wastewater with a $Cd^{2+}$ concentration of 800 mg/L was prepared, and electrodeposition was carried out in a reactor with a high gravity field. The schematic diagram of the high-gravity intensified process of electrodeposition is shown in Figure 14a. It is found that the high gravity field helps bubble detachment and mass transfer enhancement, which is beneficial to maintain the active regions on the electrode. The schematic diagram of the high-gravity intensified process of the electrodeposition experiment is shown in Figure 14b. Under the conditions of a current density of 53.1 A/m$^2$, NaCl concentration of 0.2 mol/L, initial pH of 2, wastewater circulation flow rate of 64 L/h, and electrodeposition time of 120 min, the removal rate of $Cd^{2+}$ was 99.4%, and the current efficiency was 44.8%. The Cd content in the sediment was up to 91.29%, and the main phases were metallic Cd and $Cd(OH)_2$, which can be used as the crude Cd metal in the downstream refining process [129].

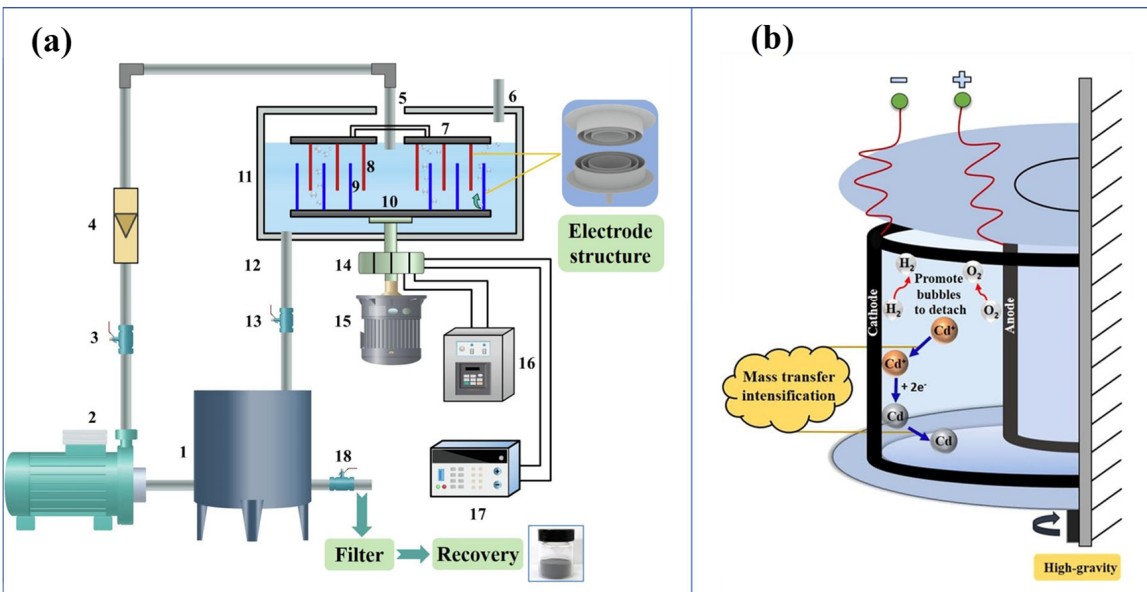

**Figure 14.** (**a**) Schematic diagram of the experimental setup for the high gravity enhanced electrodeposition process: 1—storage tank; 2—pump; 3, 13, 18—valve; 4—rotameter; 5—liquid phase inlet; 6—gas phase outlet; 7—cathode plate; 8—cathode; 9—anode; 10—anode plate; 11—MCCE-RB; 12—liquid phase outlet; 14—slip ring; 15—motor; 16—high-gravity control system; 17—DC power supply or potentiostat; (**b**) Schematic diagram of high-gravity intensified the process of electrodeposition, both reproduced from [129], with permission from Elsevier, 2022.

Electrodeposition can also be used for the separation of $Cd^{2+}$ and $Mn^{2+}$ in solution. Stainless steel and $Ti/IrO_2$ materials are used as the cathode and anode, respectively. $Cd^{2+}$ in solution was reduced to Cd at the cathode and $Mn^{2+}$ was oxidized to $MnO_2$ at the anode.

At the solution pH of 2 and current density of 370 A/m$^2$, the concentration of Mn$^{2+}$ in solution remained almost constant after reaction for 4h, while the concentration of Cd$^{2+}$ decreased from 2207 mg/L to 4.6 mg/L, and the obtained Cd product had a purity of 98% [130].

### 3.3.9. Electrocoagulation

In the electrocoagulation method, electric current is applied to dissolve the anode metal to release metal ions and oxygen bubbles, and cathode water decomposes to produce hydroxide ions and hydrogen bubbles. Metal ions and hydroxide ions react to produce electroflocculants, which can adsorb pollutants from waste solutions. The commonly used electrode materials are iron (mild steel or galvanized iron), aluminum, and stainless steel [131]. The upfloated oxygen and hydrogen bubbles transport the adsorbed pollutants to the surface of the solution, promoting the removal of pollutants from the wastewater [85]. The main reaction equations are shown below.

$$\text{Anode: } M \rightarrow M^{n+} + ne^- \quad 2H_2O \rightarrow O_2 + 4H^+ + 4e^- \tag{11}$$

$$\text{Cathode: } 2H_2O \rightarrow H_2 + 2OH^- \tag{12}$$

$$\text{Solution: } M^{n+} + nOH^- \rightarrow M(OH)_n \tag{13}$$

Interactions occurring within an electrocoagulation cell were divided into three stages, as shown in Figure 15: (1) electrolytic oxidation and in-situ formation of coagulants; (2) destabilization of contaminants, emulsions, and particulates; and (3) floc formation.

Sometimes electrocoagulation can be combined with flotation, wherein the flocculating particles are carried to the surface of the solution by externally introduced bubbles and collected in the foam. Factors affecting the efficiency of electrocoagulation in removing pollutants from wastewater include the pH of the solution, conductivity, temperature, and the configuration and condition of the electrodes [132]. The flocs formed by electrocoagulation are larger than those formed by ordinary sedimentation, which makes liquid—solid separation easier. The electrocoagulation method is still considered for low-concentration heavy metal wastewater due to the driving force of the applied electric field [133]. Electrocoagulation is environmentally friendly without secondary pollution; however, the electrodes used for electrocoagulation need to be replaced periodically due to consumption and passivation of the electrodes [134].

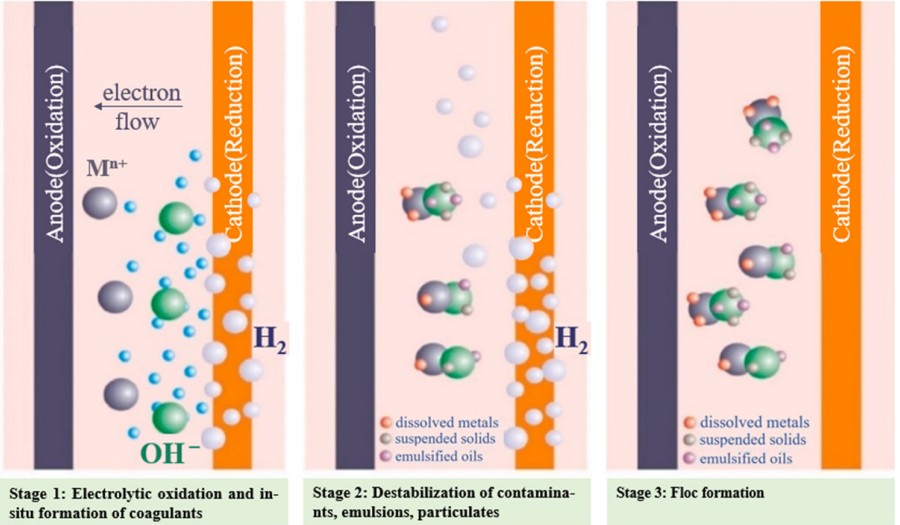

**Figure 15.** Interactions occur within an electrocoagulation cell based on the three stages, reproduced from [134], with permission from Elsevier, 2022.

The simulated Cd-containing wastewater with an initial $Cd^{2+}$ concentration of 25 mg/L was treated by electrocoagulation. The dissolved oxygen concentration was controlled in a closed atmosphere electrocoagulation system using iron as an electrode, and the generated green rust flocs had the adsorption ability for $Cd^{2+}$. Increasing the initial pH of the simulated wastewater could induce the coprecipitation of $Cd(OH)_2$ and flocs, which can dramatically improve the removal efficiency of $Cd^{2+}$. After electrocoagulation at pH of 9, the residual $Cd^{2+}$ content was 0.02 mg/L [133]. For the aluminum alloy as the electrode, the removal rate of $Cd^{2+}$ from simulated Cd-containing wastewater with an initial $Cd^{2+}$ concentration of 10–50 mg/L was 97.5% at a current density of 0.2 A/$dm^2$ and pH of 7. The $Cd^{2+}$ concentration could be reduced to less than 0.005 mg/L after treatment by electrocoagulation, and the adsorption of $Cd^{2+}$ by aluminum hydroxide in the electrocoagulation process was described by the Langmuir adsorption isotherm [135]. Moreover, acid mine wastewater in the Nicosia region of Cyprus has a $Cd^{2+}$ concentration of 0.042 mg/L and contains many heavy metal ions, such as $Fe^{3+}$, $Zn^{2+}$, $Mn^{2+}$, and $Cu^{2+}$. This mine wastewater was treated by electrocoagulation with aluminum alloy as the electrode, and the removal rate of $Cd^{2+}$ was 96%. In addition, the removal rates of other heavy metal ions were also higher than 98% [136].

3.3.10. Foam Extraction

Foam extraction includes flotation extraction, ion flotation, and precipitation flotation, and foam extraction has been demonstrated to be a promising separation technology for recovering valuable metals from dilute aqueous solutions [137–140]. Compared with conventional solvent extraction, foam extraction offers several important advantages, such as facile and flexible operation, higher mass transfer efficiency, and lower reagent consumption [141,142].

Foam extraction has been studied more extensively for the treatment of wastewater containing heavy metal ions, such as $Cu^{2+}$, $Pb^{2+}$, $Zn^{2+}$, and potential $Cd^{2+}$ [143]. Figure 16a shows the adsorption mechanism of the metal ions and surfactant to form hydrophobic particles, and the Figure 16b describes the removal and recovery of hydrophobic particles by the flotation process. Humic acid is a kind of efficient collector for the metal ions chelation. $Cu^{2+}$, $Pb^{2+}$, $Zn^{2+}$ in the wastewater can readily combine with the humic acid to form suspending precipitates, metal humic acid (shorted to MHA). In order to rapidly separate the contaminants, flocculants and surfactants would be added into the composite solution system to increase particle size and hydrophobicity. $Fe^{3+}$ and CTAB had favorable flocculation effects on the MHA due to the electrostatic adsorption and binding ability of the reactive groups (−COOH, −OH) in humic acid with the high-valent metal ions. It is testified that the size of the MHA increased from 10 to 30 mm after the addition of $Fe^{3+}$ and CTAB, and the removal of $Cu^{2+}$, $Pb^{2+}$, and $Zn^{2+}$ from the solution after froth flotation was 98.7–99.9% [144]. It is also demonstrated that the flotation extraction method was effective for the removal of $Cd^{2+}$ from the electroplating wastewater with an initial $Cd^{2+}$ concentration of 63 mg/L, and 97.39% of $Cd^{2+}$ was removed as the foam flotation was conducted at pH of 8, contact time of 60 min, sodium dodecyl sulfate (SDS) surfactant dosage of 0.2 g, and pressure of 137.89 kPa. After that, the remaining $Cd^{2+}$ concentration of the purified water was only 1.13 mg/L [145].

In addition, the xanthate and hexadecyltrimethylammonium bromide presents satisfactory $Cd^{2+}$ removal effect, and they have strong binding ability with $Cd^{2+}$. Potassium ethyl xanthate (KEtX) can easily combine with $Cd^{2+}$ to form the Cd-KEtX complexes with a low solubility, the complexation balance has a higher stability constant [137–139]. Hexadecyltrimethylammonium bromide (HDTMA) is also showing a superior collecting ability of $Cd^{2+}$. Foam extraction was then followed to remove the complexes with a high efficiency over 99%. More importantly, the conventional precipitation method is often applicative for highly concentrated solution, while the foam extraction maintained a 99% removal of $Cd^{2+}$ at the lower $Cd^{2+}$ concentration less than 0.113 mg/L [146].

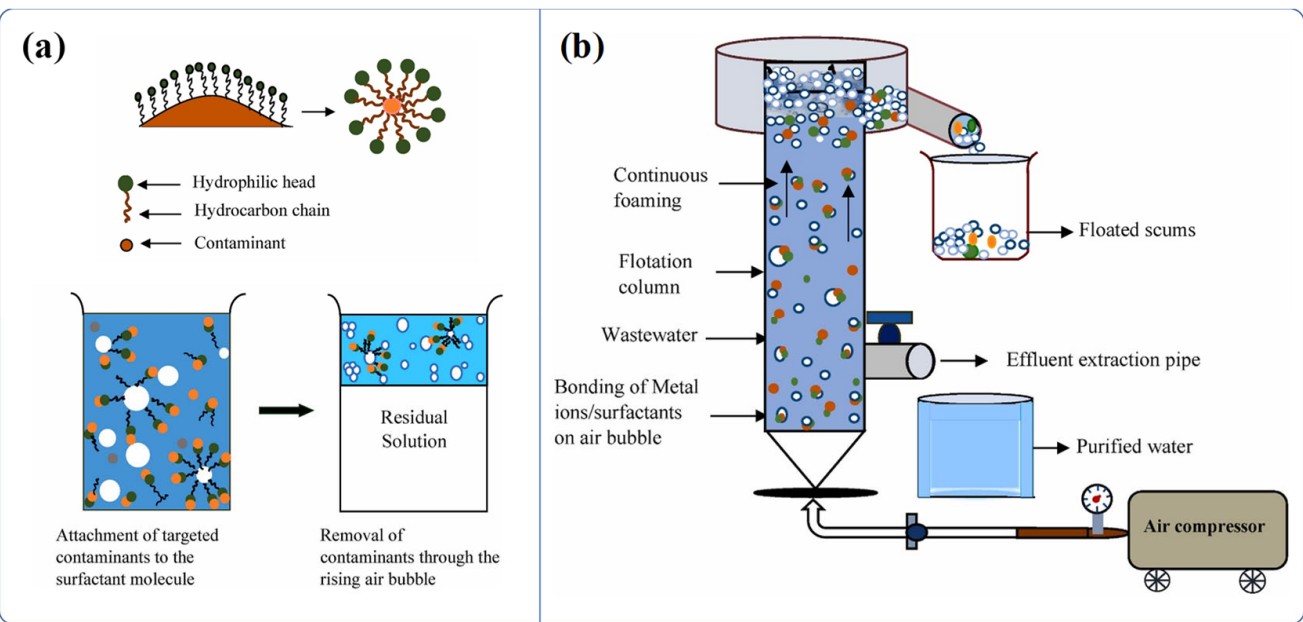

**Figure 16.** (**a**) Interaction mechanism of agents and ions; (**b**) Schematic diagram of flotation system, reproduced from [147], with permission from Elsevier, 2022.

### 3.3.11. Comparison of Different Methods

The advantages, limitations, and specific applications of the different methods for Cd removal and recovery from waste solutions are listed in Table 2. The cementation and precipitation methods are based on the chemical reaction mechanism. These processes are simple and inexpensive to operate, but the cementation method may have the disadvantage of excessive consumption of zinc powder, while the obvious shortcoming of the precipitation method lies in the further treatment of settled sludge. The solvent extraction method has a wide range of applications and high selectivity for metal ions, but the high cost of the extractant limits the large-scale application of the solvent extraction method.

The ion exchange method, adsorption method, membrane separation method, and electrodialysis method are simple and efficient with low consumption and have greater application prospects. Further research is needed for the regeneration of ion exchange resin, desorption of the adsorbent, and increasing the durability of the membrane. Electrodeposition and electrocoagulation methods use electrochemical promotion reactions to proceed, with higher product purity and less environmental pollution, but the initial cost is higher, and the electrode passivation problem would reduce the removal effect of Cd.

Foam extraction for removing heavy metal ions from solution has the advantages of a simple process, low cost, good treatment effect, high enrichment ratio for metal ions, and is also applicable at low concentration, which has good application prospects. There is still a need to optimize flotation agents, such as precipitation complexing agents, flocculants and surfactants, and to thoroughly study the mechanism of particle transport and separation at the gas—liquid-solid three-phase interface during flotation.

**Table 2.** Comparison of the advantages, limitations, and specific applications of methods for the removal and recovery of Cd from waste solutions.

| Method | Advantages | Limitations | Specific Application | Initial Content (mg/L) | Removal Rate/Adsorption Capacity | Residual Content (mg/L) | Ref. |
|---|---|---|---|---|---|---|---|
| Cementation | ✓ Simple operation<br>✓ High efficiency<br>✓ Recover pure metal | ☐ Large amount of Zn powder<br>☐ Separation of Zn and Cd is difficult | Graded addition of Zn powder | 640–740 | 99.9% | - | [80] |
| | | | Reduce the size of Zn powder | 400 | - | 1 | [83] |
| | | | Electrical enhancement | 20,000 | 99.21% | - | [79] |
| Precipitation | ✓ Simple operation<br>✓ Low cost<br>✓ Easy to control | ☐ Low removal rate<br>☐ Difficult solid—liquid separation<br>☐ Generates large amounts of sludge that needs to be treated | $Cd(OH)_2$ precipitation | 1200 | - | 0.086 | [87] |
| | | | $CdCO_3$ precipitation | 100 | 99.9% | - | [51] |
| | | | CdS precipitation | 70.9 | - | 0.08 | [88] |
| | | | $Cd_5H_2(AsO_4)_4 \cdot 4H_2O$ Coprecipitation | 22,000 | 99.7% | - | [86] |
| Ion exchange | ✓ High efficiency<br>✓ High selectivity<br>✓ Resin is renewable | ☐ High cost<br>☐ Secondary pollution during resin regeneration | Chelating resin D-401 | 675 | 245 mg/g | - | [90] |
| | | | G-26 | - | 99.68% | - | [91] |
| | | | MTS9570 | - | 98.95% | - | [91] |
| Solvent extraction | ✓ Simple operation<br>✓ Wide application<br>✓ High selectivity | ☐ High cost<br>☐ Organic reagents pollute the environment | D2EHPA | 5000 | 3% | - | [99] |
| | | | MDEHPA | 100 | 90.9% | - | [100] |
| Adsorption | ✓ High efficiency<br>✓ Rapid reaction<br>✓ Wide application<br>✓ Easy to recycle | ☐ Low selectivity<br>☐ Poor effect at high concentration | Poplar sawdust | 180 | 49.32 mg/g | - | [107] |
| | | | BCB24 | 100 | 47.39 mg/g | - | [108] |
| | | | M−rGO | 35 | 262.79 mg/g | - | [111] |
| | | | Ternary magnetic ABI composite | 250 | 219.2 mg/g | - | [112] |
| | | | SGO | 80 | 43.45 mg/g | - | [113] |
| | | | $Fe_3O_4$@PB@NTA | 25 | 310.56 mg/g | - | [115] |
| | | | ZrMOF@GSH | 200 | 393 mg/g | - | [117] |
| | | | CaFuMOF | 21.24 | 781.2 mg/g | - | [118] |
| Membrane separation | ✓ Simple operation<br>✓ Rapid reaction<br>✓ Wide application<br>✓ Low energy consumption | ☐ Low selectivity<br>☐ Membrane contamination | Ceramic-supported-polymeric composite NF membrane | 5 | 95.5% | - | [120] |
| | | | Mn-Fh/Cell/PVA composite membrane | 50 | 11.11 mg/g | - | [121] |
| | | | Poly(vinyl alcohol)/chitosan nanofiber | 400 | 148 mg/g | - | [122] |
| Electrodialysis | ✓ High efficiency<br>✓ Rapid reaction<br>✓ Effective at low concentration | ☐ High energy consumption<br>☐ Low selectivity<br>☐ Membrane contamination | BMED | - | 75.8% | - | [126] |
| | | | Complexation electrodialysis | 5.63 | - | 0.3 | [127] |
| Electrodeposition | ✓ High product purity<br>✓ High efficiency<br>✓ Low pollution to the environment | ☐ High initial cost<br>☐ Poor effect at low concentration | Provide high gravity field | 800 | 99.4% | - | [129] |
| | | | Selective electrodeposition | 2207 | - | 4.6 | [130] |
| Electrocoagulation | ✓ High efficiency<br>✓ Easy liquid—solid separation<br>✓ No secondary pollution | ☐ High cost<br>☐ The electrode needs to be replaced regularly | Green rust floc | 25 | - | 0.02 | [133] |
| | | | Alternating current | 50 | - | 0.005 | [135] |
| | | | Existence system of multiple heavy metal ions | 0.042 | 96% | - | [136] |
| Foam extraction | ✓ Rapid reaction<br>✓ Simple operation<br>✓ High separation efficiency<br>✓ Large handling capacity<br>✓ Low reagent consumption<br>✓ Applicable at low concentrations | ☐ Need to develop high-selectivity reagents | SDS surfactant | 63 | Over 98% | 1.13 | [145] |
| | | | Cd-KEtX and HDTMA | 0.113 | 99% | - | [146] |

## 4. Conclusions

Cd is a highly toxic and carcinogenic heavy metal element that is a serious hazard to both the natural environment and human health, and Cd and its compounds have been classified as Class I carcinogens by the WHO. Due to the low content and high dispersion in the Earth's crust, most Cd resources are endowed with sphalerite, galena, and chalcopyrite, and Cu-Pb-Zn smelting is the main source of industrial Cd pollution.

This paper reviews various methods for the removal and recovery of Cd from Cd-containing secondary resources, such as dust, slag, and waste solutions, generated during the Cu-Pb-Zn smelting process. At present, the removal and recovery of Cd mainly adopt hydrometallurgical processes, that is, first leaching with sulfuric acid solution or hydrogen peroxide solution to transfer $Cd^{2+}$ in dust and slag to the solution and then separating and enriching $Cd^{2+}$ in solution by various removal and recovery methods. The removal and recovery methods of $Cd^{2+}$ in solution mainly include cementation, precipitation, ion exchange, solvent extraction, adsorption, membrane, electrodialysis, electrodeposition, electrocoagulation, and foam extraction.

The various methods were compared in terms of technical principles, process parameters, and separation efficiency. The traditional chemical method, although simple and inexpensive to operate, requires the addition of many chemicals in the treatment process and reprocess the resulting waste sludge and waste liquid. Ion exchange, adsorption, and membrane methods are simple and efficient with low energy consumption, but the contamination and regeneration of the materials used need to be addressed. The consumption of electrode materials and the high cost caused by passivation in electrochemical methods limit the application of electrochemical methods. Foam extraction is a promising method to remove and recover Cd, especially for low concentration waste solutions, and the development of highly selective and low-cost foam extraction reagent is a promising and applicable research direction.

**Author Contributions:** G.H.: Writing—review & editing, Funding acquisition. J.W.: Visualization, Investigation, Writing—original draft, Data curation. H.S.: Data curation. B.L.: Conceptualization, Methodology, Writing—original draft, Funding acquisition. Y.H.: Conceptualization, Project administration, Funding acquisition. All authors have read and agreed to the published version of the manuscript.

**Funding:** The authors wish to express their thanks to the National Natural Science Foundation of China (No. U2004215, 52150079, 52174263), the Key Scientific Research Project Plan of Henan Colleges and Universities (No. 22A450001), the Natural Science Foundation of Henan Province China (No. 222300420075, 222301420030), and the Innovation Talents Support Program in University of Henan Province (No. 20HAST1T012, 23HASTIT004) for the financial support.

**Institutional Review Board Statement:** Not applicable.

**Informed Consent Statement:** Not applicable.

**Data Availability Statement:** Not applicable.

**Conflicts of Interest:** The authors declare that they have no known competing financial interests or personal relationships that could have appeared to influence the work reported in this paper.

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
