# Peer review of "A Critical Review on the Removal and Recovery of Hazardous Cd from Cd-Containing Secondary Resources in Cu-Pb-Zn Smelting Processes"

_metals, doi:10.3390/met12111846_

Round 1

Reviewer 1 Report

The paper is good enough to be accepted for publication, as it contributes to the state of the art of the treatment and disposal of cadmium-rich industrial wastes. However, no thermodynamic and kinetic results are included, so several important topics such as thermodynamic predictions from activity measurements or reaction rate of the different cited proceses could be important to identify probable attaining of poor optimization results.

Reviewer 2 Report

The present Review paper presents the recovery methods of Cd from dust, slag and waste solutions, and compares the technical principles, process parameters, separation efficiency, advantages and disadvantages, and application requirements.

The manuscript is clearly written and organized. The grammar and sentence structure are fine. In my opinion, it meets the publishing standards of “Metals” and could be published after minor revision.

Comments

1. Cd is a key component of Ni-Cd batteries and CdTe 18 semiconductor materials.

CdTe now accounts for around a 7% market share and is the first of the second-generation thin film technologies. Today, third-generation cells, although are less commercially, are advanced ‘emerging’ technologies. This includes organic photovoltaics (OPVs), copper zinc tin sulfide (CZTS), perovskite solar cells, dye-sensitized solar cells (DSSCs), and quantum dot solar cells.

2. This review can provide guidance for selecting appropriate methods to treat Cd-containing solid waste and waste solutions 24 in a harmless manner.

It could be removed

3. In addition, a new route to treat Cd-containing solutions via the foam extraction method

It should be removed (from abstract and from text)

4. China has large lead and zinc ore reserves with measured resources of approximately 95.2 million tons and 35 large Pb and Zn deposits that are larger than 1 million tons [40].

The information should not be given only for China. The Authors should mention the reserves or production all over the world, correspondingly.  

5. Cd-containing secondary resources

It should be changed, since the Authors describes the processes.

Information about the chemical analysis and the mineralogy of those secondary wastes should be added.

6. Cd-containing dust, which contains approximately 2%-25% Cd, mainly in the form of CdO or CdS.

Average chemical analysis and mineralogy of the dust should be added (in the text)

7. In recent decades, most of the slags produced by the Cu-Pb-Zn smelters were directly landfilled without harmless treatment

Average chemical analysis and mineralogy of the slag should be presented.

8. 3.3.1. Substitution

The method is called “cementation”

9. These processes are simple and inexpensive to operate, but the consumption of Zn powder and precipitant in the process as well as substitution and precipitation products would cause pollution to the natural environment

This is not true and should be removed. Cementation is the most effective method, especially in Zn electrolyte solution, since pure Cd powder can be recovered.
